# RNA structure prediction using positive and negative evolutionary information

**Elena Rivas** *

Department of Molecular and Cellular Biology, Harvard University, Cambridge, Massachusetts, USA

* elenarivas@fas.harvard.edu

## Abstract

Knowing the structure of conserved structural RNAs is important to elucidate their function and mechanism of action. However, predicting a conserved RNA structure remains unreliable, even when using a combination of thermodynamic stability and evolutionary covariation information. Here we present a method to predict a conserved RNA structure that combines the following three features. First, it uses significant covariation due to RNA structure and removes spurious covariation due to phylogeny. Second, it uses negative evolutionary information: basepairs that have variation but no significant covariation are prevented from occurring. Lastly, it uses a battery of probabilistic folding algorithms that incorporate all positive covariation into one structure. The method, named CaCoFold (Cascade variation/covariation Constrained Folding algorithm), predicts a nested structure guided by a maximal subset of positive basepairs, and recursively incorporates all remaining positive basepairs into alternative helices. The alternative helices can be compatible with the nested structure such as pseudoknots, or overlapping such as competing structures, base triplets, or other 3D non-antiparallel interactions. We present evidence that CaCoFold predictions are consistent with structures modeled from crystallography.

**Data Availability Statement:** All relevant data are within the manuscript and its Supporting information files.

**Funding:** The author received no specific funding for this work.

## Author summary

The availability of deeper comparative sequence alignments and recent advances in statistical analysis of RNA sequence covariation have made it possible to identify a reliable set of conserved base pairs, as well as a reliable set of non-basepairs (positions that vary without covarying). Predicting an overall consensus secondary structure consistent with a set of individual inferred pairs and non-pairs remains a problem. Current RNA structure prediction algorithms that predict nested secondary structures cannot use the full set of inferred covarying pairs, because covariation analysis also identifies important non-nested pairing interactions such as pseudoknots, base triples, and alternative structures. Moreover, although algorithms for incorporating negative constraints exist, negative information from covariation analysis (inferred non-pairs) has not been systematically exploited. Here I introduce an efficient approximate RNA structure prediction algorithm that incorporates all inferred pairs and excludes all non-pairs. Using this, and an improved visualization tool, I show that the method correctly identifies many non-nested structures in

**Competing interests:** The authors have declared that no competing interests exist.

agreement with known crystal structures, and improves many curated consensus secondary structure annotations in RNA sequence alignment databases.

## Introduction

Having a reliable method to determine the structure of a conserved structural RNA would be an important tool to be able to elucidate major biological mechanisms, and will open the opportunity of discovering new ones. Structure and biological function usually are closely related, as in the case of riboswitches where the structure dictates the biological function [1, 2], or the bacterial CsrB RNA which acts as a sponge to sequester the CsrA protein [3], or the 6S RNA which mimics the structure of a DNA promoter bound to the RNA polymerase to regulate transcription [4].

The importance of comparative information to improve the prediction of a conserved RNA structure has been long recognized and applied to the determination of RNA structures [5–13]. Computational methods that exploit comparative information in the form of RNA compensatory mutations from multiple sequence alignments have been shown to increase the accuracy of RNA consensus structure prediction [14–19]. Indeed, the majority (97–98%) of the ribosomal RNA (rRNA) secondary structure was inferred before the crystal structure was available using covariation analysis [20].

Several challenges remain in the determination of a conserved RNA structure using comparative analysis. There is ample evidence that pseudoknotted basepairs covary at similar levels as other basepairs, but most comparative methods for RNA structure prediction can only deal with nested structures. Identifying pseudoknotted and other non-nested pairs that covary requires having a way of measuring significant covariation due to a conserved RNA structure versus other sources.

Here, in addition to using positive information in the form of basepairs observed to significantly covary, it would also be advantageous to use negative covariation information in the form of basepairs that should be prevented from occurring because they show variation but not significant covariation. The use of both positive and negative information from covariation analysis can be compared to how chemical and enzymatic probing data is used to infer RNA structures, using propensities for modification of paired versus unpaired bases [21–25] to infer 2D [26–28] and 3D [29] RNA structure.

To approach these challenges, we have previously introduced a method called R-scape (RNA Structural Covariation Above Phylogenetic Expectation) [30] that reports basepairs that significantly covary using a tree-based null model to estimate phylogenetic covariation from simulated alignments with similar base composition and number of mutations to the given one but where the structural signal has been shuffled. Significantly covarying pairs are reported with an associated E-value describing the expected number of non-structural pairs that could have a covariation score of that magnitude or larger in a null alignment of similar size and similarity. We call these significantly covarying basepairs for a given E-value cutoff (typically $\leq 0.05$) the positive basepairs.

The covariation measure used by R-scape takes into account any possible residue mutations, and does not restrict itself to changes compatible with standard Watson-Crick base pairs or G:U wobble pairs. Any nucleotide exchange, such as A:G in a Watson-Crick base pairs, or other mutations in non Watson-Crick base pairs contribute to the covariation score. R-scape uses the G-test covariation measure, as G-test reports the best sensitivity and specificity behavior in comparison to other covariation measures [30]. G-test has the same mathematical

expression as mutual information (MI), but uses residue counts instead of frequencies. The effect of such change is that for two pairs of positions with the same MI, the pair with fewer gaps will have the larger G-test score.

In addition to reporting positive basepairs, that is, significantly covarying basepairs of any kind, R-scape has recently introduced another method to estimate the covariation power of a pair based on the mutations observed in the corresponding aligned positions [31]. Where a pair of position shows no significant covariation, this method allows distinguishing between two different cases: a pair that has too little sequence variation and may still be a conserved basepair, versus a pair with adequate sequence variation but where the variation is inconsistent with a covarying basepair. This latter case should be rejected as basepairs. We call these pairs with variation but not covariation the negative basepairs. In summary, positive basepairs that have significant covariation do not have to be consistent with standard Watson-Crick basepairs, and negative pairs that show variation but not covariation could include instances of consistent basepairs.

Here we combine these two sources of information (positive in the form of significantly covarying basepairs, and negative in the form of pairs of positions unlikely to form basepairs) into a new RNA folding algorithm. The algorithm also introduces an iterative procedure that systematically incorporates all positive basepairs into the structure while remaining computationally efficient. The recursive algorithm is able to find pseudoknots, other non-nested interactions, alternative structures and triplet interactions provided that they are supported by covariation. The algorithm also predicts additional helices without covariation support but consistent with RNA structure. Helices with covariation-supported basepairs tend to be reliable. Additional helices lacking covariation support are less reliable and need to be taken as speculative.

We use the alignments provided by the databases of structural RNAs Rfam [32] and the Zasha Weinberg Database (ZWD) [33] to produce CaCoFold structure predictions. The number of positive pairs (that is, significantly covarying basepairs proposed by R-scape) is constant for a given alignment and an annotated structure. Here we study the differences between the positive pairs incorporated into the CaCoFold structures versus those present in the annotated structures, comparing with structures derived by crystallography when possible.

## Results

### The CaCoFold algorithm

The new RNA structure prediction algorithm presents three main innovations: the proposed structure is constrained both by sequence variation as well as covariation (the negative and positive basepairs respectively); the structure can present any knotted topology, basepair of any type [34] (not just Watson-Crick), and include residues pairing to more than one residue; and all positive basepairs are always incorporated into a final RNA structure. Pseudoknots and other non-nested pairwise interactions, as well as alternative structures and tertiary interactions are all possible provided that they have covariation support.

The method is named Cascade covariation and variation Constrained Folding algorithm (CaCoFold). Despite exploring a 3D RNA structure beyond a set of nested Watson-Crick basepairs, the algorithm remains computationally tractable because it performs a cascade of probabilistic nested folding algorithms constrained such that at a given iteration, a maximal number of positive basepairs are forced into the fold, excluding all other positive basepairs as well as all negative basepairs. Each iteration of the algorithm is called a layer. The first layer calculates a nested structure that includes a maximal subset of positive basepairs. Subsequent layers of the algorithm incorporate the remaining positive basepairs arranged into alternative helices.

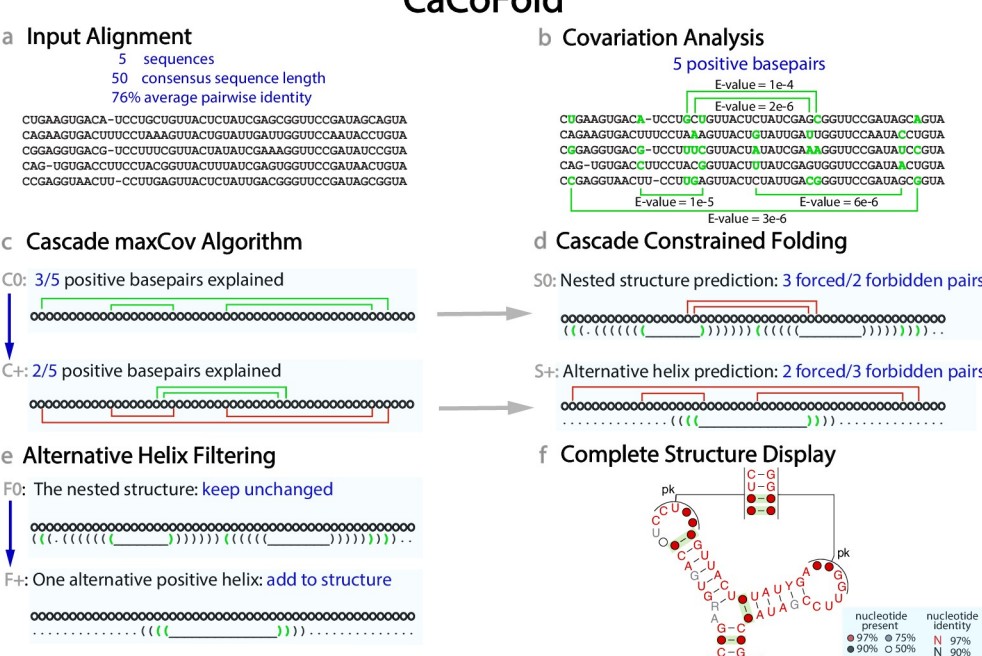

**Fig 1. The CaCoFold algorithm. (a)** Toy alignment of five sequences. **(b)** The statistical analysis identifies five significantly covarying position pairs in the alignment (E-value < 0.05). Column pairs that significantly covary are marked with green arches, compensatory pairwise substitutions including G:U pairs are marked green relative to consensus (black). **(c)** The maxCov algorithm requires two layers to explain all five covariations. In the first (C0) layer, three positive basepairs depicted in green are grouped together. In successive layers (C+), positive basepairs already taken into account (depicted in red) are excluded. **(d)** At each layer, a dynamic programming algorithm produces the most probable fold constrained by the assigned positive basepairs (green parentheses), to the exclusion of all negative basepairs and other positive basepairs (red arches). (This toy alignment does not include any negative basepairs.) Residues forming a red arch can pair to other bases. Basepairs that do not significantly covary are depicted by black parentheses. **(e)** The S+ alternative structures without positive basepairs that overlap in more that half of their residues with the S0 structure are removed. Alternative helices with positive basepairs are always kept. **(f)** The final consensus structure combining the nested S0 structure with the alternative filtered helices from all other layers is displayed automatically using a modified version of the program R2R. Positive basepairs are depicted in green.

From an input alignment, the positive basepairs are calculated using the G-test covariation measure with APC correction after removing covariation signal resulting from phylogeny, as implemented in the software R-scape [30]. G-test allows all possible nucleotide exchanges. The set of all significantly covarying basepairs is called the positive set. Positive pair can be of any type, and allow exchanges other than Watson-Crick (A:U, C:G or G:U). We also calculate the covariation power for all possible pairs [31]. The set of all pairs that have variation but not covariation is called the negative set. Negative pairs are highly variable and could include cases of Watson-Crick consistent pairs. Operationally, positive pairs have an E-value smaller than 0.05, and negative pairs are those with covariation power (the expected sensitivity of significantly covarying) larger than 0.95 and significance E-value larger than one. Non-significantly covarying pairs with an E-value between 0.05 and 1 are allowed (but not forced) to basepair regardless of power. All positive basepairs are included in the final CaCoFold structure, and all negative basepairs are forbidden to appear.

Fig 1 illustrates the CaCoFold algorithm using a toy alignment (Fig 1a) derived from the manA RNA, a structure located in the 5' UTRs of cyanobacterial genes involved in mannose metabolism [35]. After R-scape with default parameters identifies five positive basepairs

**a** **Model used by the maxCov algorithm**

### Nussinov Grammar

```
S -> o S            any non-covarying residue
S -> o S o S        a covarying basepair
S -> S S
S -> end
```

**b** **Model used by the folding algorithm (first layer)**

### RNA Basic Grammar (RBG)

```
S -> o S            a free unpaired residue
S -> L S
S -> end

L ->  o F o         a helix starts
L ->  o P o         a one-basepair helix ends

F ->  o F o         a helix adds one more basepair
F ->  o P o         a helix ends
```

what can happen at the end of a helix...

```
P ->       o...o         a hairpin     loop
P -> o...o L             a left   bulge loop
P ->       L o...o       a right bulge loop
P -> o...o L o...o       an internal   loop
P -> M1 M                a multiloop starts

M -> M1 M      multiloop adds one more branch
M -> R         multiloop about to add right residues

R -> R  o      a right-unpaired residue in multiloop
R -> M1        multiloop about to add left residues

M1 -> o M1     a left-unpaired residue in multiloop
M1 -> L        multiloop starts another helix
```

**c** **Model used by the folding algorithm (additional layers)**

### G6X Grammar

```
S -> L
S -> L S
S -> end

L -> o F o      a helix starts
L ->  o o       a basepair of contiguous residues
L ->   o        an unpaired residue

F -> o F o      a helix adds one more basepair
F ->  o o       a helix ends without a hairpin
F ->  L S       a helix ends, more stuff to come
```

```
 o           a non-covarying RNA residue
o o          a covarying RNA basepair
o            an RNA residue, not forming any basepairing
o...o        a set of contiguous unpaired RNA residues
o o          an RNA basepair; bases could be at arbitrary distance in the RNA backbone
S,L,F,P,M,M1,R   non-terminals that have to be transformed following one of the allowed rules
```

**Fig 2. RNA models used by the CaCoFold algorithm.** (**a**) The Nussinov grammar implemented by the maxCov algorithm uses the R-scape E-values of the significantly covarying pairs, and maximizes the sum of -log(E-value). (**b**) The RBG model used by the first layer of the folding algorithm. (**c**) The G6X model used by the rest of the layers completing the non-nested part of the RNA structure. For the RBG and G6X models, the F nonterminal is a shorthand for 16 different non-terminals that represent stacked basepairs. The three models are unambiguous, that is, given any nested structure, there is always one possible and unique way in which the structure can be formulated by following the rules of the grammar.

(Fig 1b), the CaCoFold algorithm calculates in four steps a structure including all five positive basepairs as follows.

**(1) The cascade maxCov algorithm.** The cascade maxCov algorithm groups all positive basepairs in nested subsets (Fig 1c). At each layer, it uses the Nussinov algorithm, one of the simplest RNA models [36]. Here we use the Nussinov algorithm not to produce an RNA structure, but to group together a maximal subset of positive basepairs that are nested relative to each other. Each subset of nested positive basepairs will be later provided to a folding dynamic programming algorithm as constraints. Fig 2 includes a detailed description of the Nussinov algorithm.

The first layer (C0) finds a maximal subset of compatible nested positive basepairs with the smallest cumulative E-value. After the first layer, if there are still positive basepairs that have not been explained because they did not fit into one nested set, a second layer (C1) of the max-Cov algorithm is performed where only the still unexplained positive basepairs are considered. The cascade continues until all positive basepairs have been grouped into nested subsets.

The cascade maxCov algorithm determines the number of layers in the algorithm. For each layer, it identifies a maximal subset of positive basepairs forced to form, as well as a set of basepairs not allowed to form. The set of forbidden basepairs in a given layer is composed of all negative pairs plus all positive pairs not in the current layer.

The cascade maxCov algorithm provides the scaffold for the full structure, which is also obtained in a cascade fashion.

**(2) The cascade folding algorithm.**   For each layer in the cascade with a set of nested positive basepairs, and another set of forbidden pairs, the CaCoFold algorithm proceeds to calculate the most probable constrained nested structure (Fig 1d).

Different layers use different folding algorithms. The first layer is meant to capture the main nested structure (S0) and uses the probabilistic RNA Basic Grammar (RBG) [37]. The RBG model features the same basic elements as the nearest-neighbor thermodynamic model of RNA stability [26, 38] such as basepair stacking, the length of the different loops, the length of the helices, the occurrence of multiloops, and others. RBG simplifies some details of loops in the models used in the standard thermodynamic packages, such as ViennaRNA [26], Mfold [39], or RNAstructure [27] resulting in fewer parameters, but it has comparable performance regarding folding accuracy [37]. Fig 2 includes a description of the RBG algorithm.

The structures at the subsequent layers (S+ = {S1, S2, ...}) are meant to capture additional helices with covariation support that do not fit into the main secondary structure S0. We expect that covariation in the subsequent layers will correspond to pseudoknots, and also non-nested tertiary contacts, or base triplets. The S+ layers add alternative helices (complementary or not) to the main secondary structure, for that reason instead of a full loop model like RBG, the S+ layers use the simpler G6 RNA model [40, 41] which mainly models the formation of helices of contiguous basepairs. Here we extend the G6 grammar to allow positive pairs that are parallel to each other in the RNA backbone, interactions that are not uncommon in RNA motifs. We name the modified grammar G6X (see Fig 2 for a description).

The RBG and G6X model parameters are trained on a large and diverse set of known RNA structures and sequences as described in Ref. 37. At each layer, the corresponding probabilistic folding algorithm reports the structure with the highest probability using a CYK dynamic programming algorithm on a profile sequence that contains information on the proportion of each nucleic acid in each consensus column of the alignment.

Because the positive residues that are forced to pair at a given cascade layer could pair (but to different residues) at subsequent layers, the CaCoFold algorithm can also identify triplets or higher order interactions (a residue that pairs to more than one other residue) as well as alternative helices that may be incompatible and overlap with other helices.

**(3) Filtering of alternative helices.**   In order to combine the structures found in each layer into a complete RNA structure, the S+ structural motifs are filtered to remove redundancies without covariation support.

We first break the S+ structures into individual alternative helices. A helix is operationally defined as a set of contiguous basepairs. Basepairs disrupted by just a one or two residue bulge or by a 1x1 internal loop are still considered one single helix. Under this operational definition, a helix can consist of just one basepair. This is a slight misuse of the word "helix" which strictly speaking involves a spatial rotation and a translation between two basepairs. The algorithm makes sure that each basepair belongs to one and only one helix. A helix is arbitrarily called positive if it includes at least one positive basepair.

All positive alternative helices are reported. Alternative helices without any covariation support are reported only if they include at least 15 basepairs, and if they overlap in no more than 50% of the bases with another helix already selected from previous layers. This is an empirically determined cutoff that applies only to alternative helices without any covariation support.

In our simple toy example, there is just one alternative helix. The alternative helix is positive, and it is added to the final structure. No helices are filtered out in this example (Fig 1d).

**(4) Automatic display of the complete structure.** The filtered alternative helices are reported together with the main nested structure as the final RNA structure. We adapted the program R2R to visualize the CaCoFold structure with all covarying basepairs annotated in green [42]. CaCoFold reports and draws a consensus structure for the alignment. Conserved positions display the residue identity color coded by conservation (red > 97%, black > 90%, and gray > 75%), otherwise a circle is displayed colored by column occupancy (red > 97%, black > 90%, gray > 75%, white > 50%).

The CaCoFold drawings of the (presented or predicted) consensus structures depict all nested and non-nested basepairs automatically (Fig 1f). For presented 3D structures, non Watson-Crick basepairs (regardless of whether they overlap or not) are annotated as non-canonical ("nc"), and Watson-Crick alternative helices as pseudoknots ("pk"). For a predicted CaCoFold structure, alternative helices are annotated as pseudoknots ("pk") if they do not overlap with the main nested structure, or as triplets ("tr") if they overlap. Thus, a CaCoFold alternative helix labeled "tr" includes at least one pair in which one the residues is also involved in a different pair. CaCoFold cannot distinguish whether an alternative helix labeled "tr" is a triplet or part of an alternative structure. Similarly, a CaCoFold alternative helix labeled as "pk" could also be part of an alternative fold, and it does not necessarily correspond to a helix composed only of Watson-Crick basepairs.

If R-scape does not identify any positive basepair, one single layer is defined without positive pairs and constrained only by the negative pairs, and one nested structure is calculated. Lack of positive basepairs results in a proposed structure without evolutionary support.

For the toy example in Fig 1, R-scape with default parameters identifies five positive basepairs. The CaCoFold algorithm requires two layers to complete. The first layer incorporates three nested positive basepairs. The second layer introduces the remaining two positive basepairs. The RBG fold with three constrained positive basepairs produces three helices. The G6X fold with two positive and three forbidden basepairs results in one alternative helix between the two hairpin loops of the main nested structure. In this small alignment there are no negative basepairs, and no alternative helices without covariation support have to be filtered out. The final structure is the joint set of the four helices, and includes one pseudoknot.

## CaCoFold finds pseudoknots, triplets and other long and short-range interactions

For a realistic example of how CaCoFold works, we present in Fig 3 an analysis of transfer-messenger RNA (tmRNA). The tmRNA is a bacterial RNA responsible for freeing ribosomes stalled at mRNAs without a stop codon. The tmRNA molecule includes a tRNA-like structural domain, and a mRNA domain which ends with a stop codon. The tmRNA molecule is typically 230-400 nts [43], and its proposed structure includes a total of 12 helices forming four pseudoknots [44]. The core elements of the tmRNA structure are well understood, but the molecule has a lot of flexibility and is thought to undergo large conformational changes with the 4 pseudoknots forming a ring around a part of the small subunit of the ribosome [43].

We performed the analysis on the tmRNA seed alignment in Rfam (RF0023) which includes 477 sequences. The length of the consensus sequence is 354 nts, and the average pairwise percentage identity is 42% (Fig 3a). In step one, the covariation analysis on the input alignment results on 121 significantly covarying basepairs (Fig 3b). This analysis ignores the proposed Rfam consensus structure and performs the covariation analysis on the alignment alone. This result is in agreement with the covariation power estimated for this alignment,

# transfer-messenger RNA (tmRNA)

## a  Input Alignment
Rfam RF00023 seed alignment
- 477  sequences
- 354  consensus sequence length
- 357  average    sequence length
- 42% average pairwise identity

## c  Cascade maxCov Algorithm
121 positive basepairs explained in 6 layers

| | |
|---|---|
| layer 1: 69 | layer 2: 41 |
| layer 3: 5 | layer 4: 3 |
| layer 5: 2 | layer 6: 1 |

## e  Alternative Helix Filtering

18 alternative helices
- 5 pseudoknots
- 3 triplets
- 10 mRNA-induced covariations

## b  Covariation Analysis
All possible pairs analyzed equally
119 annotated basepairs in alignment
(not used in analysis)
414 columns analyzed:
- 121 positive basepairs (significantly covary)
- 109 positive basepairs expected by power
- 31,027 negative basepairs

## d  Cascade Constrained Folding
- 139 annotated pairwise interactions
- 121/139 positive  basepairs
- 74 pairs not in final ss due to forbidden negative basepairs

## f  Complete structure  display

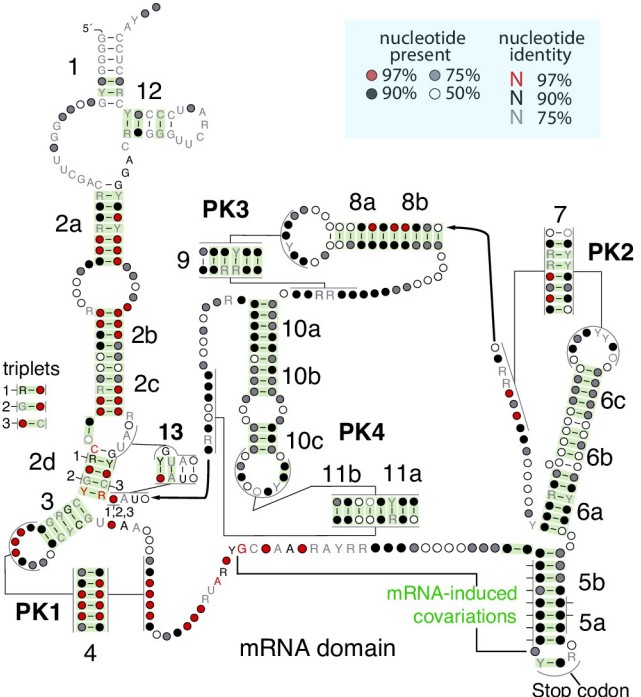

## g  Structure comparison
Kelley *et al.*, RNA 2001, Fig 4

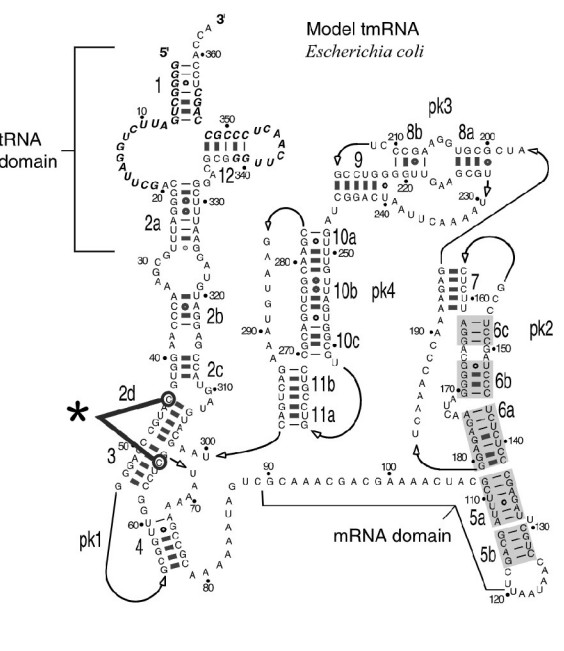

**Fig 3. The CaCoFold algorithm applied to the transfer-messenger RNA (tmRNA).** Steps **(a)** to **(f)** refer to the same methods as described in Fig 1. **(a)** Characteristics of the input alignment. **(b)** The statistical test that considers all possible pairs equally resulting in the assignment of 121 significantly covarying positive basepairs. The Rfam consensus structure in not used in the analysis. The whole analysis is performed using the single command `R-scape --fold` on the input alignment. The analysis takes 25 seconds (30s including drawing all the figures) on a 3.3 GHz Intel Core i7 MacBook Pro. **(c)** The maxCov algorithm requires 6 layers to incorporate all 121 positive basepairs. **(d)** The cascade Constraint folding completes the structure with a total of 139 basepairs. **(e)** After filtering, there are five pseudoknotted helices, three triplets and 10 other mRNA-induced covariations. The structural display in **(f)** has been modified by hand to match the standard depiction of the tmRNA secondary structure in **(g)**. The thick line in (g) marked with an asterisk indicates the C-C triplet interaction proposed in Ref. 44. Details of the mRNA-induced covariations are given in S6(c) Fig.

which expects to find on average 109 significantly covarying basepairs [31]. In the next step, the maxCov algorithm requires 6 layers to explain all 121 positive basepairs (Fig 3c). Next, the constrained folding of each of the 6 layers results on a total of 139 annotated pairwise interactions.

The covariation analysis also identifies 31,027 negative pairs (out of a total of 85,491 possible pairs for 414 columns analyzed), those are forbidden to form because they show variation but not covariation. In the final structure, 74 basepairs are not reported do to the forbidden negative pairs (Fig 3d). The final alternative helix filtering step reports: 5 pseudoknots, 3 triplets and 10 other covariations that are induced by coding constraints, which we describe in more detail below. All alternative helices have covariation support (Fig 3e).

The CaCoFold structure for the tmRNA is given in Fig 3f, and it includes the 12 helices and four pseudoknots [44]. It also proposes an additional helix (H13) with covariation support. We have not identified H13 in tmRNA crystal structures. Due to the amount of overlap between H13 and helix H2d, this could indicate the presence of two alternative competing structures.

In the helix H2d/helix H3 region, CaCoFold identifies three triplets, one of them (triplet 1) is confirmed by the structure derived from the tmRNA EM structure (13.6 Å) with PDB ID 3IZ4 [45]. A different triplet for which we do not find covariation signal has been previously proposed in that same region (marked with an asterisk in Fig 3g). This is a complex region with many 3D contacts as helix H2d interacts both with the PK1 and PK4 pseudoknots [43].

CaCoFold identified 10 additional interactions associated to the mRNA domain. These tend to occur between contiguous residues. These interactions are not related to the RNA structure and arise from coding constraints (more details in S6(c) Fig). We observe this kind of covariation in other coding mRNA regions, not just in tmRNA. Finally, CaCoFold reports one covariation between the first and second position of the stop codon in the mRNA domain. The U residue in the first position of the stop codon is invariant, so a covariation involving this reside should not occur. This spurious covariation arises from a misalignment of the stop codon in the Rfam seed alignment. A small rearrangement of the alignment in that region results in a conserved stop codon.

We compared the tmRNA CaCoFold structure to the structure predicted for the same alignment by RNAalifold, a ViennaRNA program for predicting a consensus structure [26]. S1(a) Fig shows the output of RNAalifold. RNAalifold does not predict pseudoknots or any other non-nested structure, and it only identifies 6 of the 12 helices in the tmRNA structure (Fig 3g). RNAalifold predicts 46 basepairs, but it does not assign confidence to the proposed basepairs. In S1(b) Fig, the covariation analysis of the tmRNA alignment shows that 45 of the 46 RNAalifold basepairs covary. But it also indicates that there are 76 other covarying basepairs not present in the RNAalifold structure (Fig 3b). CaCoFold brings together the basepair validation provided by the covariation analysis with a structure that incorporates all 121 inferred basepairs.

## RNAs with structures improved by positive and negative signals

We have produced CaCoFold structures from the alignments provided by the databases of structural conserved RNAs Rfam [32] and ZWD [33]. Unlike the previous section where the proposed consensus structure was ignored, here we perform two independent covariation tests: one on the set of basepairs in the annotated consensus structure, another on the set of all other possible pairs (option "to improve an existing structure" in Methods). It is important to notice that because of this two-set analysis, CaCoFold builds on the knowledge provided by the alignments and the consensus structures of Rfam and ZWD. Using the positive and

negative pairs obtained from the covariation test as constraints, the CaCoFold structure is then built anew.

One strength of the CaCoFold algorithm is that the structure is guided by significant covariation, and the final structure is viewed in the light of all covarying basepairs. For alignment with little or no significant covariation, CaCoFold behaves as the RBG model, which we have shown in benchmarks perform similarly to standard methods [37]. Because in the absence of covariation RNA structure prediction lack reliability and all methods perform comparably, we concentrate on the set of RNAs with covariation support.

Another strength of the CaCoFold algorithm is in incorporating all covariation signal present in the alignment into one structure. When the CaCoFold structure includes the same covarying pairs than the structure in the database, the differences between the two structures can only occur in regions not reliably predicted by either of the methods, thus we concentrate on the set of RNAs for which the CaCoFold structure has different covariation support than the annotated structure.

Because the set of positive pairs is constant and CaCoFold incorporates all of them, CaCoFold structures cannot have fewer positive pairs than the database consensus structures. Here we investigate the set of RNAs with CaCoFold structures with different (that is, more) covariation support than the annotated structures, and whether those differences are consistent with experimentally-determined 3D structures when available.

We identify 277 (out of 3,030) Rfam families and 105 (out of 415) ZWD RNA families for which the CaCoFold structure includes positive basepairs not present in the given structures. Because there is overlap between the two databases, in combination there is a total of 313 structural RNAs for which the CaCoFold structure has more covariation support than either the Rfam structure or the ZWD structure. Of the 314 RNAs, there are five for which the Rfam and ZWD alignments and structures are different (PhotoRC-II/RF01717, manA/RF01745, radC/RF01754, pemK/RF02913, Mu-gpT-DE/RF03012) and we include both versions in our analysis. In the end, we identify a total of 319 structural alignments for which the structure presented in the databases is missing positive basepairs, and CaCoFold proposes a different structure with more covariation support. In Table 1, we classify all structural differences into 15 types.

## CaCoFold structures consistent with 3D structures

The set of 319 CaCoFold structures with more covariation support includes 27 RNAs that have 3D structures for representative sequences (out of a total of 97 families with 3D structures). We tested that for those RNAs (21 total, leaving aside 1 LSU and 5 SSU rRNA) the CaCoFold structure predictions are indeed supported by the 3D structures. Table 2 describes the 21 RNAs: 5S RNA, tRNA, 6S RNA, Group-II intron, two bacterial RNase P RNAs (A-type and B-type), tmRNA, two SRP RNAs (bacterial and archaeal), four snRNAs (U2, U3, U4, and U5), and eight riboswitches (FMN, SAM-I, Cobalamin, Fluoride, Glutamine, cyclic di-AMP, and YkoK leader). The comparison of the CaCoFold structures for those 21 RNAs to 3D structures are presented in Figs 4 and 5 and supplemental S2–S6 Figs.

In Fig 4a, we show the *RNase P RNA A-type* where CaCoFold identifies the two pseudoknots, one of which (P6) is not in the Rfam consensus structure. RNase P RNA A-type also includes five long-range tertiary interactions [46, 68] highlighted in black and numbered "1" to "5" in Fig 4a. CaCoFold identifies covarying basepairs in two of those regions: a Y-R covarying pair ("tr_1") in the P8/P14-hairpin-loop region "3", and a R-R covarying pair ("tr_2") in the P8/P18-hairpin-loop region "4".

**Table 1. CaCoFold structures with different covariation support than the structures provided with the structural alignments.** CaCoFold structures with different covariation support can only have more positive basepairs. **(Left)** The 319 structural RNAs (from the Rfam and ZWD databases combined) for which the CaCoFold structure has more covariation support are manually classified into 15 categories. Each RNA is assigned to one main type, although they can belong to others as well. Examples of types 1-11 are presented in S7 Fig. A full description of all 319 RNAs is given in the supplemental table.

| | Modifications introduced by the extra covariations in the CaCoFold structure | # RNAs |
|---|---|---|
| Type 1 | Helix extended by additional covariations | 23 |
| Type 2 | New helix with covariation support | 12 |
| Type 3 | One helix completely modified | 7 |
| Type 4 | New pseudoknot with covariation support | 16 |
| Type 5 | New junction/internal loop or coaxial stacking | 17 |
| Type 6 | Internal loop/multiloop reshaped by coaxial stacking | 12 |
| Type 7 | Hairpin/internal loop covariations (often nonWC) | 19 |
| Type 8 | Non Watson-Crick (not within a loop) covariations | 24 |
| Type 9 | Base triples | 28 |
| Type 10 | (Cross,Side)-covariations (see text) | 30 |
| Type 11 | Possible alternative structures | 6 |
| Type 12 | Additional covariations in SSU and LSU rRNA | 6 |
| Type 13 | Covariations not supporting a secondary structure | 3 |
| Type 14 | Misalignment introducing spurious covariations | 2 |
| Type 15 | Low power; inconclusive | 114 |
| CaCoFold structures with different (*i.e.* more) covariation support | | 319 |

Fig 4b shows the *SAM-I riboswitch* where CaCoFold identifies the reported pseudoknot [47]. Other RNAs for which CaCoFold identifies unannotated pseudoknots with covariation support confirmed by crystallography include five riboswitches: the *Cobalamin riboswitch* [49] (Fig 5a), *FMN riboswitch* [58] S3(b) Fig, *ZMP/ZTP riboswitch* [59] S3(c) Fig, *Fluoride riboswitch* [60] S4(a) Fig, *Glutamine riboswitch* [61] S4(b) Fig, and the *Archaeal SRP RNA* [62] S4(c) Fig.

Also in the SAM-I riboswitch, CaCoFold identifies an apparent lone Watson-Crick A-U pair in the junction of the four helices which in fact stacks with helix P1 [47].

The SAM-I riboswitch [47] CaCoFold structure also includes additional covariations that further extend existing helices P2a, P3 and P4. Other RNAs for which CaCoFold identifies additional covarying pairs in helices supported by 3D structures are given in S2 Fig: *Bacterial SRP RNA* [54] S2(a) Fig, *cyclic di-AMP riboswitch* [55] S2(b) Fig, and *YkoK leader* [56] S2(b) Fig.

In Fig 4c, the *U4 spliceosomal RNA* shows two covarying pairs (labeled "1" and "3" in the figure) which define one additional helix relative to the Rfam structure, and help identify a kink turn RNA motif. Both the helix and the kink turn motif are confirmed by the crystal structure [48].

Four other examples of RNAs for which CaCoFold identifies key covarying residues missing in the Rfam structure that define important structural elements are described next. For *RNaseP B-type* in S5(a) Fig, one covarying basepair (labeled "1" in blue in the figure) in helix P15.1 identifies a new internal loop in the CaCoFold structure relative to the Rfam structure that is confirmed in the crystal structure [63].

For the *Group-II intron* fragment in S5b Fig, one additional R-Y covarying basepair (labeled "1" in blue) defines a three-way junction not present in the Rfam structure, and confirmed by the crystal structure [64].

**Table 2. 21 RNAs with 3D structures and CaCoFold structures with different covariation support than the structures provided with the structural alignments.** Subset of 21/319 CaCoFold structures with more covariation support for which there is 3D structural information (not including the 6 rRNAs). We compare the 21 CaCoFold predicted structures to the 3D structures in Figs 4, 5 and Supplemental S2–S6 Figs. The associated "types" are described in Table 1.

| 21/319 RNAs with 3D structures | | | |
|---|---|---|---|
| RNA | Rfam seed alignment | Types | Figure |
| RNase P RNA A-type [46] | RF00010 | 4,8 | Fig 4a |
| SAM-I riboswitch [47] | RF00162 | 1,4,6 | Fig 4b |
| U4 snRNA [48] | RF00015 | 2,5 | Fig 4c |
| Cobalamin riboswitch [49] | RF00174 | 1,4,5 | Fig 5a |
| tRNA [50, 51] | RF00005 | 1,8,9 | Fig 5b |
| U2 snRNA [52, 53] | RF00004 | 11 | Fig 5c |
| Bacterial SRP RNA [54] | RF00169 | 1 | S2(a) Fig |
| cyclic di-AMP riboswitch [55] | RF00379 | 1 | S2(b) Fig |
| YkoK leader [56] | RF00380 | 1 | S2(c) Fig |
| 5S rRNA [57] | RF00001 | 3,5 | S3(a) Fig |
| FMN riboswitch [58] | RF00050 | 1,4 | S3(b) Fig |
| ZPM-ZTP riboswitch [59] | RF01750 | 4,9 | S3(c) Fig |
| Fluoride riboswitch [60] | RF01734 | 1,4 | S4(a) Fig |
| Glutamine riboswitch [61] | RF01739 | 4 | S4(b) Fig |
| Archaea SRP RNA [62] | RF01857 | 4 | S4(c) Fig |
| RNase P RNA B-type [63] | RF00011 | 5 | S5(a) Fig |
| Group-II intron (fragment) [64] | RF02001 | 5 | S5(b) Fig |
| U5 snRNA [65] | RF00020 | 5,7,10 | S5(c) Fig |
| Fungal U3 snoRNA [66] | RF01846 | 5 | S6(a) Fig |
| 6S RNA [4] | RF00013 | 8 | S6(b) Fig |
| tmRNA [67] | RF00023 | 9,10,11,14 | S6(c) Fig |

For *U5 snRNA* [65] in S5(c) Fig, there is a new Y-Y covarying pair that modifies a hairpin loop. This non Watson-Crick interaction consists of a C:U pair exchanging to a U:C pair in 46% of the sequences, and it is annotated by the software RNAView as a stacked pair [69]. Stacked pairs do not belong to any of the major families of base pair types but are important in forming the RNA structure [69], and as a result they also covary.

For the *Fungal U3 snoRNA* S6(a) Fig, an additional R-Y covarying pair in the CaCoFold structure relative to the Rfam annotation (labeled with "1" in S6(a) Fig) allows to identify the characteristic boxB/boxC boxes of the snoRNA [66].

In Fig 5a, the CaCoFold structure for the Cobalamin riboswitch [49] includes a pseudoknot, a covarying pair identifying a multiloop with coaxial stacking, and additional covarying base-pair in helices P1 and P2 all supported by the 3D structure [49]. CaCoFold also identifies other unreported covarying pairs in the internal loop between helices P7 and P8.

The tRNA CaCoFold structure derived from the Rfam seed alignment is given in Fig 5b. The variable loop (V loop) forms an additional helix only in type II tRNAs, but not in type I tRNAs [70]. About 15% of the tRNAs in the Rfam seed alignment are of type II, but the V loop basepairs are not annotated in the consensus structure. Under default parameters, R-scape does not analyzed the V loop positions as the number of gaps is above the cutoff of 75% gaps. By allowing the analysis of all positions, we observe three Watson-Crick basepairs in the V loop.

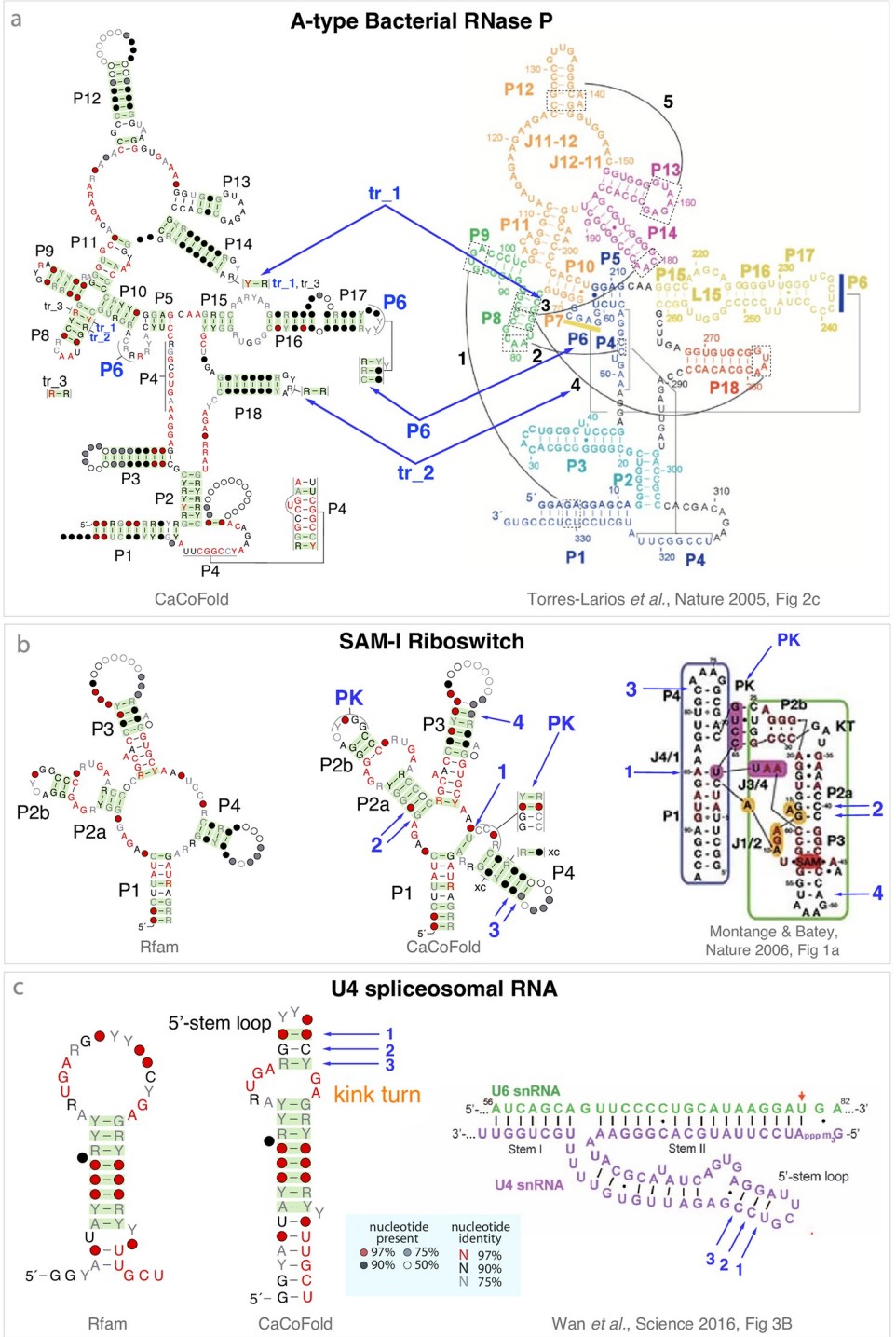

**Fig 4. CaCoFold structures confirmed by known 3D structures (part 1/7).** Structural elements with covariation support introduced by CaCoFold relative to the Rfam annotation and corroborated by 3D structures are annotated in blue. **(a)** The A-type RNase P RNA CaCoFold structure includes relative to the Rfam structure one more helix (P6) and two significant covariations, named tr_1 and tr_2. Blue arrows show the placement of these three covarying motifs relative to the 3D structure [46]. The display of the crystal structure has been modified to indicate with back shaded boxes five regions with tertiary interactions labeled "1" to "5"[68]. "tr_1" occurs in region "3" between P8 and the hairpin loop of P14, and "tr_2" in region "4" representing the interaction between P8 and the hairpin loop of P18. The display of the CaCoFold structure has been modified by hand to match the standard depiction of the structure. **(b)** The SAM-I riboswitch CaCoFold structure shows relative to the Rfam structure one more helix forming a pseudoknot, and

a A-U pair stacking on helix P1 both confirmed by the SAM-I riboswitch 2.9 Å resolution crystal structure of *T. tengcongensis* [47]. CaCoFold also identifies additional pairs with covariation support for helices P2a, P3 and P4. **(c)** The U4 snRNA CaCoFold structure identifies one more internal loop and one more helix than the Rfam structure confirmed by the 3D structure [48]. The new U4 internal loop flanked by covarying Watson-Crick basepairs includes a kink turn (UAG-AG). The non Watson-Crick pairs in a kink turn (A-G, G-A) are generally conserved (> 97% in this alignment) and do not covary.

The tRNA CaCoFold structure also incorporates thirteen other long-range interactions not in the Rfam tRNA structure. We have used the crystal structure 1EHZ of the yeast phenylalanine type I tRNA at 1.93 Å resolution [51], and the RNAView annotation [69] of all its basepairs to produce a 3D-derived annotation of the consensus structure in the Rfam alignment. Five of the CaCoFold long-range interactions (depicted in blue) are confirmed by the 3D-derived structure, and for clarity, in Fig 5b they have been renamed to match the 3D structure.

CaCoFold identifies six additional covarying pairs not reported by RNAView on the 1EHZ tRNA crystal structure. The positive basepair (marked "1") in the anticodon hairpin is a non-canonical basepairs that has been confirmed as a single hydrogen-bonded pair [71]. The "pk", "pk_1" and "pk_2" interactions occur amongst the same residues. "pk_1" and "pk_2" have worse E-values than "pk" (0.007 and 0.00002 compared to $1e^{-14}$) and may be secondary covariations between the D and T loops. The "tr_2" triplet has been reported as specific to type 1 tRNAs (base triplet "g" in Ref. [70]). There are four additional CaCoFold covarying pair between backbone adjacent residues that happen in the three hairpin loops. tRNA pseudo-genes, likely to be present in the Rfam seed alignment, may have induce spurious covariation in this analysis.

Finally, two CaCoFold covarying pairs (depicted in orange) in the anticondon (AC) loop appear to be the result of constraints other than RNA structure. One involves one anticodon residue and the discriminator residue in the acceptor stem. This anticodon/discriminator covariation results from the interaction of both residues with aminoacyl-tRNA synthetase [50].

In Fig 5c, the U2 spliceosomal snRNA describes a case of alternative structures. "Stem IIc" was originally proposed as possibly forming a pseudoknot with one side of Stem IIa, but was later discarded as non-essential for U2 function [52, 72]. But later, a U2 conformational switch was identified indicating that Stem IIa and Stem IIc do not form a pseudoknot but are two competing helices promoting distinct splicing steps [53]. Both helices are important to the U2 function, and both have covariation support.

5S rRNA S3(a) Fig shows the case of a region (the helix 4 and Loop E region) almost completely reshaped by the covariations found by CaCoFold, and in agreement with the 3D structure [57].

In addition to the coding mRNA signal in tmRNA S6(c) Fig, we have found another signal that produces non-phylogenetic covariations in the 6S RNA S6(b) Fig which regulates transcription by direct binding to the RNA polymerase [4]. The 6S RNA structure mimics an open promoter and serves as a transcription template. Synthesis of a 13 nt product RNA from the 6S RNA results in a structural change that releases the RNA polymerase. We do not find any covariation evidence for the alternative helix of "isoform 2" in Ref. 4 S6(b) Fig, but we observe one covariation between the U initiating the RNA product and the previous position. We hypothesize an interaction of the two bases with the RNA polymerase.

## Other CaCoFold structures with more covariation support

Based on what we learned from the 3D structures, we manually classified the 319 RNAs with modified structures into 15 categories (Table 1). In Supplemental S1 Table, we report a full list

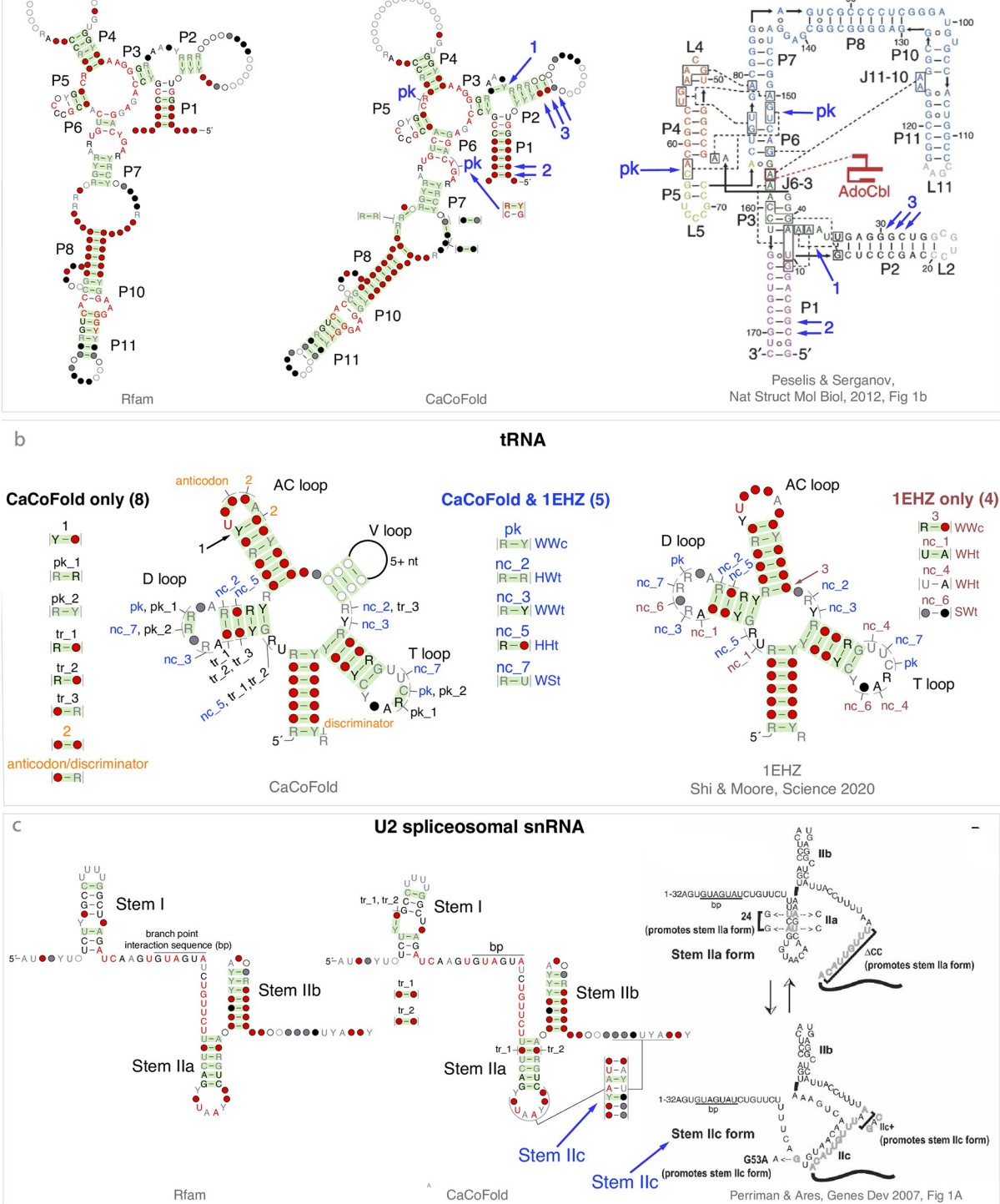

**Fig 5. CaCoFold structures confirmed by known 3D structures (part 2/7).** Structural elements with covariation support introduced by CaCoFold relative to the Rfam annotation and corroborated by 3D structures are annotated in blue. **(a)** Relative to the Rfam structure, the Cobalamin riboswitch CaCoFold structure adds one pseudoknot and one Watson-Crick basepair defining a four-way junction between helices P1, P2, and P3, both confirmed by the *S. thermophilum* crystal structure [49]. It also adds more covariation support for helices P1 and P2. **(b)** In CaCoFold structures, alternative helices that do not overlap with the nested structure are annotated as pseudoknots (pk), otherwise they are annotated as triplets (tr). For structures obtained from a crystal structure, non Watson-Crick basepairs are annotated as non-canonical (nc)

regardless of whether they are overlapping or not with the nested structure. The tRNA CaCoFold structure has been re-annotated manually to match the labeling of the *S. cerevisiae* phenylalanine tRNA 1EHZ crystal structure (1.93 Å) for all common basepairs [51]. Of the covarying pairs in the CaCoFold structure but not in the Rfam tRNA structure, five (depicted in blue) are confirmed by the 1EHZ structure as analyzed by RNAView. The sequence of the 1EHZ tRNA does not include the V loop, which appears in 16% of the 954 sequences in the Rfam tRNA seed alignment. Two covarying pairs (depicted in orange) appear to be the result of constraints other than RNA structure. The remaining six covarying pairs are labeled in black. Four basepairs identified in the 3D structure but not incorporated in the CaCoFold structure are depicted in brown. The annotation of the non Watson-Crick pairs with at least two H-bonds follows the nomenclature of [34] that reports the two edges of the nucleotides involved in the plain of the H-bonds. "W" stands for the Watson-Crick edge, "S" for the Sugar edge, and "H" for the Hoogsteen face; "c" and "t" stand for cis and trans respectively. WWc is a standard Watson-Crick basepairs. **(c)** In the U2 spliceosomal RNA, Stem IIa and Stem IIc, both with covariation support, are two alternative helices that compete to promote different splicing steps [53].

of the RNA families and alignments with CaCoFold structures incorporating more positive covariation support, classified according to Table 1. In S7 Fig, we show representative examples of Types 1-12 amongst the RNAs with more covariation support but without 3D structures.

In **Type 1**, the extra positive basepairs incorporated by CaCoFold extend the length of an already annotated helix, as in the *TwoAYGGAY RNA* (RF01731) and *drum RNA* (RF02958) examples. **Type 2** includes cases in which several positive basepairs identify a new helix. We present the case of the *Coronavirus 3'UTR pseudoknot*, a pseudoknotted structure specific to coronaviruses, typically 54-62 nts in length found within the 3' UTR of the N gene. The alignment for this RNA in the Rfam 14.2 Coronavirus special release (RF00165) has a consensus sequence of 62 nts, and it annotates two helices forming a pseudoknot [73]. The CaCoFold structure includes one additional third helix with 2 positive pairs and compatible with the pseudoknot. The existing chemical modification data for the Coronavirus 3'UTR pseudoknot does not rule out the presence of this additional helix [73]. **Type 3** includes seven cases in which a helix without positive basepairs in the given structure gets refolded by CaCoFold into a different helix that includes several positive basepairs. For the RF03068 *RT-3 RNA* example, the original helix has no covariation support but the refolded helix has 8 positive basepairs. **Type 4** describes cases in which positive basepairs reveal a new helix forming a pseudoknot. There are 16 of these cases, of which chrB RNA is an example. **Type 5** and **Type 6** are cases in which the additional positive basepairs refine the secondary structure, either by introducing new junctions (three-way or higher) or new internal loops (**Type 5**) or by adding positive basepairs at critical positions at the end of helices that help identify coaxial stacking (**Type 6**). For **Type 6**, we show a positive basepair in the *DUF38000-IX RNA* that highlights the coaxial stacking of two helices. **Type 7** describes cases in which the extra positive basepairs are in loops (hairpin or internal). **Types 5, 6** and **7** often identify recurrent RNA motifs [74], as in the case shown in S7 Fig, where an additional positive basepair identifies a tandem GA motif in the *RtT RNA*. Other more general non-Watson-Crick interactions are collected in **Type 8**, of which tRNA is an exceptional example where many non-Watson-Crick basepairs covary (see Fig 5(b)). The tRNA genes are highly variable in sequence, but they have a very conserved structure, and only a few positions do not covary, such as a more than 97% conserved U in the AC loop (depicted in red in Fig 5b). In S7 Fig we show another example of **Type 8**, *Bacteroides-2* a putative RNA gene identified in the Bacteriodes genus [35], where CaCoFold finds three residues involved in three basepairs. **Type 9** are putative base triplets involved in more than one positive interaction. In general, one of the positive basepairs is part of an extended helix, but the other is in general not nested and involves only one or two contiguous pairs. **Type 10** includes a particular type of base triplet that we name cross-covariation and side-covariation. A cross(side)-covariation appears when two covarying basepairs $i - j$ and $i' - j'$ that belong to the same helix are such that two of the four residues form another covarying interaction. If the extra covarying pair involves residues in one side of the helix ($i - i'$ or $j - j'$), we name it a side-

covariation (annotated "sc" in the graphical representation). If the residues are in opposite sides of the helix ($i - j'$ or $j - i'$), it is a cross-covariation (annotated "xc"). We have observed side covariations in tmRNA (Fig 3 and S6(c) Fig) and other mRNA sequences. In S7 Fig, we show an example of a helix with four cross-covariations. As an extreme example, the bacterial LOOT RNA with approximately 43 basepairs in six helices includes 28 cases of cross-covariations. **Type 11** includes a few cases in which an alternative positive helix is incompatible with another positive helix. These cases are candidates for possible competing structures. The SSU and LSU ribosomal RNA alignments are collected in **Type 12**. These are large structures with deep alignments in which about one third of the basepairs are positive. For the LSU rRNA, CaCoFold finds between 8 (Eukarya) to 22 (bacteria) additional positive basepairs. **Type 13** include just three cases for which the positive basepairs are few and cannot provide confirmation of the proposed structure. **Type 14** identifies two cases in which the Rfam and ZWD alignments report different sets of positive basepairs. These suggest the possibility of a misalignment resulting in spurious covariations. Finally, **Type 15** collects about a third (114/319) of the alignments for which CaCoFold identifies only one or two positive basepairs while the original structure has none. None of these alignments has enough covariation to support any particular structure. These alignments also have low power of covariation to decide whether there is a conserved RNA structure in the first place.

The R-scape covariation analysis and CaCoFold structure prediction including pseudoknots for all 3,016 seed alignments in Rfam 14.1 (which includes four SSU and three LSU rRNA alignments; ranging in size from SSU rRNA Archaea with 1,958 positions to LSU rRNA Eukarya with 8,395 positions) takes a total of 724 minutes performed serially on a 3.3 GHz Intel Core i7 MacBook Pro.

## Discussion

The CaCoFold folding algorithm provides a comprehensive description and visualization of all the significantly covarying pairs (even if not nested or overlapping) in the context of the most likely complete RNA structure compatible with all of them. This allows an at-a-glance direct way of assessing which parts of the RNA structure are well determined (*i.e.* supported by significant covariation). The strength and key features of the CaCoFold algorithm are in building RNA structures anchored both by all positive (significant covariation) and negative (variation in the absence of covariation) information provided by the alignment. In addition, CaCoFold provides a set of compatible basepairs obtained by constrained probabilistic folding. The set of compatible pairs is only indicative of a possible completion of the structure. They do not provide any additional evidence about the presence of a conserved structure, and some of them could be erroneous as it is easy to predict consistent RNA basepairs even from random sequences.

CaCoFold is not the first method to use covariation information to infer RNA structures [14–19], but it is the first to our knowledge to distinguish structural covariation from that of phylogenetic nature, which is key to eliminate confounding covariation noise. CaCoFold is also the first method to our knowledge to use negative evolutionary information to discard unlikely basepairs. CaCoFold differs from previous approaches in four main respects: (1) It uses the structural covariation information provided by R-scape which removes phylogenetic confounding. The specificity of R-scape is controlled by an E-value cutoff. (2) It uses the variation information (covariation power) to identify negative basepairs that are not allowed to form. (3) It uses a recursive algorithm that incorporates all positive basepairs even those that do not form nested structures, or involve positions already forming other basepairs. The CaCoFold algorithm uses different probabilistic folding algorithms at the different layers. (4)

A visualization tool derived from R2R that incorporates all interactions and highlights the positive basepairs.

Overall, we have identified over two hundred RNAs for which CaCoFold finds new significantly covarying structural elements not present in curated databases of structural RNAs. For the 21 RNAs in that set with 3D information (leaving aside SSU and LSU rRNAs), we have shown that the new CaCoFold elements are generally supported by the crystal structures. Those new elements include new and re-shaped helices, basepairs involved in coaxial stacking, new pseudoknots, long-range contacts and base triplets. Reliable CaCoFold predictions could accelerate the discovery of still unknown biological mechanisms without having to wait for a crystal structure.

We have found interesting cases of significantly covarying pairs where the covariation is not due to RNA structure, the tRNA acceptor/discriminator covariation (Fig 5b) or the coding covariations associated to the messenger domain of tmRNA (Fig 3, S6(c) Fig) are examples. These covariations do not interfere with the determination of the RNA structure, which usually forms during the first layers of the algorithm, as they are added by higher layers on top of the RNA structure. The CaCoFold visual display of all layered interactions permits to identify the RNA structure and to asses its covariation support, and may help proposing hypotheses about the origin of other interactions of different nature.

Even for RNAs with a known crystal structure, because the experimental structure may have only captured one conformation, CaCoFold can provide a complementary analysis, as in the case of the U2 spliceosomal snRNA presented here (Fig 5c). (Riboswitches also have alternative structures, but because Rfam alignments do not typically include riboswitch expression platform regions, we do not observe the alternatively structured regions of riboswitches in these data.).

CaCoFold improves the state of the art for accurate structural prediction for the many structural RNAs still lacking a crystal structure. This work provides a new tool for several lines of research such as: the study of significant covariation signatures of no phylogenetic origin present in messenger RNA, as those identified here in the tmRNA (Fig 3, S6(c) Fig); the study of the nature and origin of covariation in protein sequences; and the use of variation and covariation information to improve the quality of RNA structural alignments.

## Materials and methods

### Implementation

The CaCoFold algorithm has been implemented as part of the R-scape software package. For a given input alignment, there are two main modes to predict a CaCoFold structure using R-scape covariation analysis as follows,

- To predict a new structure: `R-scape --fold`
  All possible pairs are analyzed equally in one single covariation test. This option is most appropriate for obtaining a new consensus structure prediction based on covariation analysis in the absence of a proposed structure.
  The structures in Figs 1, 3 were obtained using this option.

- To improve a existing structure: `R-scape -s --fold`
  This option requires that the input alignment has a proposed consensus structure annotation. Two independent covariation tests are performed, one on the set of proposed base pairs, the other on all other possible pairs. The CaCoFold structure is built anew using the positive and negative basepairs as constraints.
  The structures in Figs 4, 5 and Supplemental S2–S7 Figs were obtained using this option.

### Extracting the RNA structure from a PDB file

The software is capable of obtaining the RNA structure from a PDB file for a sequence homo-log to but not necessarily represented in the alignment, and transforms it to a consensus struc-ture for the alignment.

For a given PDB [75] file, we use the software nhmmer [76] to evaluate whether the PDB sequence is homologous to the aligned sequences. If the PDB sequence is found to be a homo-log of the sequences in the input alignment, we extract the RNA structure from the PDB file (Watson-Crick and also non-canonical basepairs and contacts) using the program RNAView [69]. An Infernal model is built using the PDB sequence and the PDB-derived RNAView struc-ture [77]. All input sequences are then aligned to the Infernal PDB structural model. The new alignment includes the PDB sequence with the PDB structure as its consensus structure. We use the mapping of each sequence to the PDB sequence in this new alignment to transfer the PDB structure to the sequence as it appears in the input alignment. From all individual struc-tures, we calculate a PDB-derived consensus structure for the input alignment. The R-scape software can then analyze the covariation associated with the PDB structure mapped to the input alignment.

For example, the PDB structure and covariation analysis in Fig 5b for the tRNA (RF00005) was derived from the PDB file 1EHZ (chain A) using the options:

```
R-scape -s --pdb 1ehz.pdb --pdbchain A --onlypdb RF00005.seed.
sto
```

The option `--pdbchain <chain_name>` forces to use only the chain of name `<chain_name>`. By default, all sequence chains in the PDB file are tested to find those with homology to the input alignment. The option `--onlypdb` ignores the alignment consensus structure. By default, the PDB structure and the alignment consensus structure (if one is pro-vided) would be combined into one annotation.

### Availability

A R-scape web server is available from rivaslab.org. The source code can be downloaded from a link on that page. A link to a preprint version of this manuscript with all supplemental infor-mation and the R-scape code is also available from that page.

This work uses R-scape version 1.5.4. The distribution of R-scape v1.5.4 includes external programs: FastTree version 2.1.10 [78], Infernal 1.1.2 [77], hmmer 3.3 [79]. It also includes modified versions of the programs RNAView [69], and R2R version 1.0.6.1-49-g7bb81fb [42]. The R-scape git repository is at https://github.com/EddyRivasLab/R-scape.

For this manuscript, we used the databases Rfam version 14.1 (http://rfam.xfam.org/), the 10 new families and 4 revised families in Rfam 14.2, and ZWD (114e95ddbeb0) downloaded on February 11, 2019 (https://bitbucket.org/zashaw/zashaweinbergdata/). We used program RNAalifold from the ViennaRNA-2.4.12 software package [26].

All alignments used in the manuscript are provided in the supplemental S1 File.

### Supporting information

**S1 Fig. tmRNA structure predicted by RNAalifold and covariation analysis. (a)** The RNAa-lifold predicted consensus structure output for the tmRNA Rfam seed alignment (RF00023) obtained using default parameters. The RNAalifold structure consists of 46 basepairs, and it annotates (at least partially) 6 of the 12 helices in the structure [44]: 2 (a,b,d), 3, 5, 6, 9, and 10 (a,b,c), see Fig 3g. **(b)** The covariation analysis of the RNAalifold structure indicates that 45 of the 46 RNAalifold basepairs have covariation support (shown in green). It also identifies 76 other basepairs with covariation support not in the proposed RNAalifold structure. The

display of all 121 positive pairs can be seen in Fig 3f. (Columns with more than 75% gaps have been removed from the display.).
(PDF)

**S2 Fig. CaCoFold structures confirmed by known 3D structures (part 3/7).** Structural elements with covariation support introduced by CaCoFold relative to the Rfam annotation and corroborated by 3D structures are annotated in blue. The three RNAs are examples of CaCoFold structures with more covariation support in the form of more positive basepairs to helices already present in the consensus Rfam structures. **(a)** *SRP RNA*. The SRP complex 2XXA PDB X-ray diffraction structure has 3.94 Å resolution [54]. The PDB-derived consensus structure was obtained as described in Methods. **(b)** For the *cyclic di-AMP riboswitch*, the region around helix P4 is highly variable in the Rfam alignment, and none of the proposed structures has covariation support. The displayed CaCoFold structure showing helix P4 was obtained using a consensus reference sequence (instead of the default profile sequence). The rest of the structure has covariation support and remains invariant. **(c)** For the *YkoK leader*, there are two additional basepairs labeled "1" and "2" in helices P5 and P6 respectively confirmed by the crystal structure [56].
(PDF)

**S3 Fig. CaCoFold structures confirmed by known 3D structures (part 4/7).** Structural elements with covariation support introduced by CaCoFold relative to the Rfam annotation and corroborated by 3D structures are annotated in blue. **(a)** The *5S rRNA* CaCoFold structure remodels Helix 4 (six basepairs) and Loop C (two basepairs) in agreement with the crystal structure [57]. A Y-R covarying basepair in Loop B is not described in the 3D structure. **(b)** The *FMN riboswitch* CaCoFold structure identifies a confirmed 2-basepair pseudoknotted helix, and one covarying pair in helix P2 that is different than in the 3D structure [58]. **(c)** The covarying pseudoknot identified by CaCoFold in the *ZPM-ZTP riboswitch* is confirmed by the *Fusobacterium ulcerans* X-ray diffraction structure (2.82 Å) [59].
(PDF)

**S4 Fig. CaCoFold structures confirmed by known 3D structures (part 5/7).** Structural elements with covariation support introduced by CaCoFold relative to the Rfam annotation and corroborated by 3D structures are annotated in blue. All three cases **(a)** *Fluoride riboswitch* **(b)** *Glutamine riboswitch* **(c)** *Archeal SRP* are examples of CaCoFold structures with more covariation support in the form of a new helix forming a pseudoknot all confirmed by the 3D structures.
(PDF)

**S5 Fig. CaCoFold structures confirmed by known 3D structures (part 6/7).** Structural elements with covariation support introduced by CaCoFold relative to the Rfam annotation and corroborated by 3D structures are annotated in blue. **(a)** An additional covarying pair introduces a new internal loop in the *B-type RNase P RNA* confirmed by Ref. 63, Fig 4a. **(b)** An additional covarying pair introduces a new three-way junction in the *Group-II intron* D1D4-3 fragment [64]. **(c)** In the *U5 snRNA*, an additional Y-Y covarying pair that modifies a hairpin loop is confirmed by the *S. pombe* spliceosomal RNA cryo-EM structure 3JB9 (3.60 Å) [65].
(PDF)

**S6 Fig. CaCoFold structures confirmed by known 3D structures (part 7/7).** Structural elements with covariation support introduced by CaCoFold relative to the Rfam annotation and corroborated by 3D structures are annotated in blue. **(a)** The *U3 snoRNA* CaCoFold structure adds a covarying pair closing the boxB/boxC of the snoRNA [65]. **(b)** *6S RNA* covarying pair at the RNA synthesis initiation site not associated to RNA structure [4]. **(c)** Side-covariation in

the *mRNA-like domain* of tmRNA not due to RNA structure.
(PDF)

**S7 Fig. Examples of RNAs without a 3D structure for which the CaCoFold structure has more positive basepairs (green shading) than the structure given by the corresponding database.** We provide examples of differences corresponding to Types 1 to 11. A description of all different types is given in Table 1.
(PDF)

**S1 File. Supplemental material.** Includes alignments, and source code.
(GZ)

**S1 Table.**
(PDF)

## Acknowledgments

I thank the Centro de Ciencias de Benasque Pedro Pascual in Benasque, Spain, where ideas for this manuscript were developed. Special thanks to Ioanna Kalvari and Anton Petrov for their assistance with the Rfam database, Zasha Weinberg for assistance with the database ZWD and the program R2R, and Sean R. Eddy for comments.

## Author Contributions

**Conceptualization:** Elena Rivas.

**Data curation:** Elena Rivas.

**Formal analysis:** Elena Rivas.

**Funding acquisition:** Elena Rivas.

**Investigation:** Elena Rivas.

**Methodology:** Elena Rivas.

**Project administration:** Elena Rivas.

**Resources:** Elena Rivas.

**Software:** Elena Rivas.

**Validation:** Elena Rivas.

**Visualization:** Elena Rivas.

**Writing – original draft:** Elena Rivas.

**Writing – review & editing:** Elena Rivas.

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
