## [Decision Letter · Decision Letter 0]

11 Aug 2020

Dear Dr Rivas,

Thank you very much for submitting your manuscript "RNA structure prediction using positive and negative evolutionary information" for consideration at PLOS Computational Biology.

As with all papers reviewed by the journal, your manuscript was reviewed by members of the editorial board and by several independent reviewers. In light of the reviews (below this email), we would like to invite the resubmission of a significantly-revised version that takes into account the reviewers' comments.

We apologize for the delay in handling your manuscript but one reviewer was extremely late and we have decided to proceed without that review.

From the two reviews of your manuscript, your contribution is clearly considered important and innovative. However, both reviewers pose questions and issues on the technical aspects that should be clarified in the paper and algorithm.

Please address them in full in your revision.

We cannot make any decision about publication until we have seen the revised manuscript and your response to the reviewers' comments. Your revised manuscript is also likely to be sent to reviewers for further evaluation.

Sincerely,

Tamar Schlick

Associate Editor

PLOS Computational Biology

Nir Ben-Tal

Deputy Editor

PLOS Computational Biology

Reviewer's Responses to Questions

**Comments to the Authors:**

Reviewer #1: This is an important paper in the difficult pursuit of finding structural elements in sequence alignments. Sequence alignments are extremely powerful for establishing 2D helical regions maintained by complementary Watson-Crick pairs with GU wobbles (and the 2D structures for all “ancillary” RNAs were deduced in this way). This is based on covariations or exchanges, during evolution, between the structurally similar Watson-Crick pairs. RNAs with sequences highly conserved throughout phylogeny form therefore a difficult case. Also, the extraction of information about bases that form 3D pairs is delicate because they generally are highly conserved and do not covary strictly speaking. This has been achieved with good alignments displaying enough sequence variation. For large structures, this can be reached especially when coupled with real space molecular assembly and experimental data (mutations, chemical probing,…) in order to restrict the search range for covariations (as in JMB 216,585(90), Science 254, 853 (91), JMB 279, 773 (98)). Here, the author exploits astutely sophisticated statistical analysis to obtain reliable 2D structures with potential 3D information.

There is here original thinking and new programming yielding new data. An immense effort has been made to compare with many available RNA structures. An amazing amount of work and the results are all available in a variety of files with the codes. The paper is a real step forward, but it is hard to read and follow, not only because of the many cases discussed (too often maybe too cursorily). Some sentences state an observation but without a figure or enough precision to locate the observation in the structure. A thorough re-reading by the author would certainly help. I will give some examples below. Because of the importance of the work, I feel it is worth this additional effort to clarify some points. Some may sound trivial, but the recent controversy with long ncRNAs revealed the deep misunderstandings even among (computer) scientists working on RNA.

At the onset, the author should state clearly what nucleotide exchanges are considered to indicate covariations. Only those pertaining to standard WC pairs and GU wobble? Some GA pairs do form pairs through their Watson-Crick edges. But AG can also form “sheared “pairs (trans Hoogsteen-Sugar edge) in which case AG is not structurally the same as GA, but such AG pairs do exchange with AA pairs interacting through the same edges. Or is any variation accepted? I do not think so from reading the text. This would be useful for understanding, for example lines 38-39.

I do not understand line 146.

Line 148. I understand the definition is “operational” but a one base pair helix is not meaningful since a helix is characterized by a twist angle between steps and the rise between two steps along the helical axis.

Lines 167-168. I find the use of “triplets” as confusing. Sometimes they are used to compare with triples in 3D and sometimes with alternative structures. It is written that “nc” is used for 3D structures, but in Fig.5 for the tRNA Cacofold, one finds “nc”. That figure would need more explanations. For some nucleotides, there are several “tr” (e.g. R9) and one “nc”. See also the legend of Fig.5 (where I do not understand the second sentence): “Four nc pairs and one pk pair with covariation support are found by CaCoFold and confirmed by the 1EHZ structure. Four base triplets (tr) and two pseudoknots (pk) have covariation support but have not been assigned to any basepair type by RNAView. » See also line 331, what does this mean? Only pairs with two H-bonds are reported by RNAview and many other similar programs (for example, as discussed below 32…38 form a single H-bond contact, at most).

Fig. 5b. Among the possible positive pairs, those that fit the 3D structure are selected, but without knowing it, all those would be equivalent. I guess, this is the same for the pairs between the invariant G18G19 and invariant U55C56 where all possibilities are open and cannot be distinguished (pk1, pk2). I do not remember seeing a tRNA with U56 by the way. What is very interesting (and this is why clarity and explanations are required) is that variations between regions are observed without those variations reflecting a pair or a contact but instead a concerted variation due to tRNA/protein interactions (like anticodon triplet and N73, see Lines 327…) or global constraints on the highly functional anticodon loop (tr_4 between 36 and 38). Nucleotides 32 and 38 form a single H-bond contact and the most frequent ones are C32_A38 or Y32_Y38 (and it is one or the other depending on the anticodon triplet). For the “triplets”, nc_2 and nc_5, is it really meant as a triple or as an alternative fold? With a change in the definition of a helix this may be clarified. All tRNAs were selected, but the type I and type II tRNAs do not have the same triples. Re-doing the calculations separately for each class might enhance the values (at least for type I).

Lines 172-173. I am not sure that the first part of that sentence is correct (the second yes).

Overall, the comparisons with experimental data are not easy to follow. For the tmRNAs, there are more recent structures (one you cite somewhere but do not compare with Ramrath et al. Nature 2012; and a more recent one, Scient Rep (2018) 8:13587).

Line 217. Difficult to see that.

Line 294. Difficult to see. You may want to look at the results from comparative analysis on RNAseP, A and B (JMB 279, 773 (98)).

Lines 302, 303. What is the meaning of “??”?

Lines 314-315. State which pairs with numbers and show them clearly on the Figures.

Line 317. “Y-Y covarying pair”, which covariations are observed?

Line 339. ??

Lines 381. What is precisely meant here? Is it in part due to the huge amount of tRNA sequences and the expected changes occurring in the helices? The ‘almost’ is interesting: which ones are not involved in “covariations”?

On Fig. S7 I did not see Type 12.

Reviewer #2: The Rivas manuscript describes a new software tool for predicting

consensus RNA secondary from aligned nucleotide sequence. The tool has

a number of novel aspects that make this a useful addition to the

literature on the subject.

The approach evaluates the significance of potential covariation

signals, using a null model that accounts for phylogenetic structure

of aligned sequences. This pairs are used as constraints for iterative

rounds of Nussinov-style style structure prediction that maximises the

number of (covarying) basepairs. The iterative rounds of prediciton

mean that pseudoknots, and even alternative structures and/or triples

are captured. Further rounds of basepair prediction extend stems that

are conserved, yet don't have strong signatures of covariation.

This is a significant improvement of the state-of-the-art methods that

generally ignore pseudoknots and alternative structures -- and uses an

arguably more reasonable set of metrics for optimisation than

combining conservation, covariation and thermodynamic models.

I do think some improvements in the manuscript could be made.

1. It is probably worth pointing out that the majority (97–98%) of the

rRNA secondary structure had been inferred before the crystal

structure was published thanks to covariation analysis [1].

2. It was unclear to me what was meant by "negative information". The

MS defines this as basepairs that "show variation but not significant

covariation". So, does this include column pairs that include "U:A &

U:G" type of variation, that admittedly is not strictly covarying, but

may be structurally consistent in the right context. Some careful

definitions of positive and negative information would help the reader

immensly.

3. The use of a -log(E-value) scores concerns me slightly. With BLAST

E-values this doesn't have a very good history (e.g. https://genomebiology.biomedcentral.com/articles/10.1186/s13059-015-0721-2),

as directly using bit-scores can be a more accurate measure. I wonder if the

author considered experimenting with directly using the G-score for the first layer?

4. I'm curious about the types of Rfam families that could be improved

by CaCoFold. Were these predominantly riboswitches and ribozymes, or

other types? It's disappointing the the 5S rRNA has not been updated

in Rfam. A number of papers have now pointed out flaws in this

e.g. Rivas, Clements & Eddy (2017) and Gardner & Eldai (2015). Have

the curators of Rfam been alerted to this issue?

5. Do the improved consensus structures improve the performance of the

corresponding covariance models for homology search? Or are these

robust to errors in alignments and structures?

6. Regarding the use of PDB structures. Presumably the consensus

structure is generally a subset of basepairs parsed from 3D

structures. I did see this explicitly mentioned in the MS.

REFERENCES:

[1] Gutell, R. R., Lee, J. C., & Cannone, J. J. (2002). The accuracy

of ribosomal RNA comparative structure models. Current opinion in

structural biology, 12(3), 301-310.

Minor corrections:

A "Y-Y" covariation is mention on pg 15. Presumambly this is a

non-canonical interaction?

The "??" throughout the manuscript suggest missing references?

Figure 3: "Transfer-messenger RNA" spelled incorrectly

Installing the supplement:

# tar zxvf rscape_v1.5.2.tar.gz

<lots of="" output="">

gzip: stdin: unexpected end of file

rscape_v1.5.2/lib/hmmer/easel/documentation/figures/gev_density.pdf

rscape_v1.5.2/lib/hmmer/easel/documentation/figures/rnaseP-ecoli.pdf

tar: Unexpected EOF in archive

tar: Unexpected EOF in archive

tar: Error is not recoverable: exiting now

# cd rscape_v1.5.2/

# ./configure

<lots of="" output="">

config.status: creating src/Makefile

config.status: creating lib/FastTree/src/Makefile

config.status: creating lib/R2R/R2R-current/src/Makefile

config.status: error: cannot find input file: `lib/RNAVIEW/src/Makefile.in'

v1.5.6 seems to install fine though.</lots></lots>

**Have all data underlying the figures and results presented in the manuscript been provided?**

Reviewer #1: Yes

Reviewer #2: Yes

PLOS authors have the option to publish the peer review history of their article (what does this mean?). If published, this will include your full peer review and any attached files.

Reviewer #1: No

Reviewer #2: No
---

## [Decision Letter · Decision Letter 1]

10 Sep 2020

Dear Dr Rivas,

Thank you very much for submitting your manuscript "RNA structure prediction using positive and negative evolutionary information" for consideration at PLOS Computational Biology. As with all papers reviewed by the journal, your manuscript was reviewed by members of the editorial board and by several independent reviewers. The reviewers appreciated the attention to an important topic. Based on the reviews, we are likely to accept this manuscript for publication, providing that you modify the manuscript according to the review recommendations.

The reviewers have appreciated the thorough comments and changes made to the original manuscript,

but some technical issues remain, and one reviewer clarifies his prior comments.

Please address these points in full in a revision.

Sincerely,

Tamar Schlick

Associate Editor

PLOS Computational Biology

Nir Ben-Tal

Deputy Editor

PLOS Computational Biology

[LINK]

The reviewers have appreciated the thorough comments and changes made to the original manuscript,

but some technical issues remain, and one reviewer clarifies his prior comments.

Please address these points in full in a revision.

Reviewer's Responses to Questions

**Comments to the Authors:**

Reviewer #1: See attachment

Reviewer #2: My comments have been thoroughly addressed.

Reviewer #3: The author did not understand my first comment. SHAPE produces flexibility signals which are translated in secondary structure prediction programs by propensities of nucleotides to participate or not in base pairs; therefore equivalent to pairs and non-pairs. These programs always find acceptable solutions, and therefore they do not represent the bottleneck of RNA structure prediction. Besides, some of these programs also predict tertiary interactions (not referred to by the author). However, I agree with the author that a structure thus obtained rather corresponds to the structure under experimental observation than a structure supported by covariations.

In my initial comment on X-ray crystal structures, I said “denatured”, but what I meant is more “not under physiological conditions”. I am sorry for this lack of precision. Therefore, I do not agree with the author that "crystallographers carefully reproduce native conditions". I would rather say that the crystallographers, although they try to reproduce the native conditions, aim at obtaining crystals, whether the physiological conditions are respected or not and they seldom are. The case of U2 spliceosomal RNA is interesting because the aligned homologs lead to interactions that differ from the crystal. Q1. Does the covariation data substantiate both alternatives, or one of them only?

Now, about using RNAStructure rather than RNAalifold: I understand that it is not necessary to show both if indeed it leads to the same conclusions. Q2. Is that the case ?

Otherwise, I find that the new version is improved from the original version.

**Have all data underlying the figures and results presented in the manuscript been provided?**

Reviewer #1: Yes

Reviewer #2: Yes

Reviewer #3: Yes

PLOS authors have the option to publish the peer review history of their article (what does this mean?). If published, this will include your full peer review and any attached files.

Reviewer #1: No

Reviewer #2: No

Reviewer #3: **Yes: **Francois Major
---

## [Editor Report · Decision Letter 2]

24 Sep 2020

Dear Dr Rivas,

We are pleased to inform you that your manuscript 'RNA structure prediction using positive and negative evolutionary information' has been provisionally accepted for publication in PLOS Computational Biology.

Best regards,

Tamar Schlick

Associate Editor

PLOS Computational Biology

Nir Ben-Tal

Deputy Editor

PLOS Computational Biology

---

## [Editor Report · Acceptance letter]

19 Oct 2020

PCOMPBIOL-D-20-00868R2 

RNA structure prediction using positive and negative evolutionary information

Dear Dr Rivas,

I am pleased to inform you that your manuscript has been formally accepted for publication in PLOS Computational Biology. Your manuscript is now with our production department and you will be notified of the publication date in due course.

With kind regards,

Laura Mallard
